# Advances in Boron Neutron Capture Therapy (BNCT) for Recurrent Intracranial Meningioma

**DOI:** 10.3390/ijms24054978

**Published:** 2023-03-04

**Authors:** Tien-Li Lan, Chun-Fu Lin, Yi-Yen Lee, Ko-Han Lin, Feng-Chi Chang, Shih-Chieh Lin, Jia-Cheng Lee, Fong-In Chou, Jinn-Jer Peir, Hong-Ming Liu, Pei-Fan Mu, Yi-Wei Chen

**Affiliations:** 1Department of Heavy Particles and Radiation Oncology, Taipei Veterans General Hospital, No.201, Sec. 2, Shipai Rd., Beitou Dist., Taipei City 11217, Taiwan; 2Division of General Neurosurgery, Taipei Veterans General Hospital Neurosurgical Department, No.201, Sec. 2, Shipai Rd., Beitou Dist., Taipei City 11217, Taiwan; 3Department of Nuclear Medicinel, Taipei Veterans General Hospital, No.201, Sec. 2, Shipai Rd., Beitou Dist., Taipei City 11217, Taiwan; 4Division of Neuroradiology, Department of Radiology, Taipei Veterans General Hospital, No.201, Sec. 2, Shipai Rd., Beitou Dist., Taipei City 11217, Taiwan; 5Department of Pathology and Laboratory Medicine, Taipei Veterans General Hospital, No.201, Sec. 2, Shipai Rd., Beitou Dist., Taipei City 11217, Taiwan; 6Institute of Nuclear Engineering and Science, National Tsing Hua University, No.101, Sec 2, Kuang-Fu Rd., Hsinchu 300044, Taiwan; 7Institute of Clinical Nursing, National Yang Ming Chiao Tung University, No.155, Sec. 2, Linong St. Beitou Dist., Taipei City 112304, Taiwan

**Keywords:** Boron Neutron Capture Therapy (BNCT), targeted radiotherapy, atypical meningioma

## Abstract

Meningiomas are the most frequently diagnosed primary intracranial tumors in adults. Surgical resection is preferred if the meningioma is accessible; for those that are not suitable for surgical resection, radiotherapy should be considered to improve local tumor control. However, recurrent meningiomas are challenging to treat, as the recurrent tumor might be located in the previously irradiated area. Boron Neutron Capture Therapy (BNCT) is a highly selective radiotherapy modality in which the cytotoxic effect focuses mainly on cells with increased uptake of boron-containing drugs. In this article, we describe four patients with recurrent meningiomas treated with BNCT in Taiwan. The mean boron-containing drug tumor-to-normal tissue uptake ratio was 4.125, and the tumor mean dose was 29.414 GyE, received via BNCT. The treatment response showed two stable diseases, one partial response, and one complete response. We also introduce and support the effectiveness and safety of BNCT as an alternative salvage treatment for recurrent meningiomas.

## 1. Introduction

Meningiomas are the most frequently diagnosed primary intracranial tumors, accounting for more than 30% of all primary brain and central nervous system tumors in adult patients [1]. Although most meningiomas are benign and have a slow growth rate, some groups of meningiomas have relatively aggressive behavior and are classified as atypical meningiomas (WHO grade 2) or anaplastic meningiomas (WHO grade 3) [2]. For benign meningiomas (WHO grade 1), management decisions focus on the risks and benefits of neurological deficits and tumor control [3]. 

Thus, observation is recommended if meningiomas are accidentally discovered, have a small tumor size, and are asymptomatic. Surgical resection or radiotherapy may be considered when meningiomas start to grow rapidly or when symptoms initiate.

Compared to low-grade benign meningiomas, atypical meningiomas and anaplastic meningiomas are more aggressive and active management is usually essential [4]. Based on the location, surgical resection is preferred if the meningioma is accessible, as complete resection of the meningioma and dural attachment are independent prognostic factors for progression-free survival and overall survival based on the Simpson grading classification [5]. For patients who are not suitable for surgical resection or for whom complete resection cannot be achieved, radiotherapy should be considered to improve local tumor control [6].

However, recurrent meningiomas are challenging to treat, as the recurrent tumor might be located in the previously irradiated area, leading to the limitation of further aggressive radical radiation dosage. Boron Neutron Capture Therapy (BNCT) is a highly selective radiotherapy modality in which the cytotoxic effect focuses mainly on cells that uptake boron-containing drugs [7]. The rationale of this “targeting radiotherapy” is the discrepancy in boron-containing drug uptake by the tumor and the surrounding normal tissues [8,9]. Currently, the most widely used boron-containing drug is 4-^10^B-borono-L-phenylalanine (L-BPA), which has a chemical structure similar to that of L-tyrosine [10]. As several malignant tumors contain L-Type Amino Acid Transporter 1 (LAT-1) on the cell membrane, a relatively high uptake of L-BPA compared to that in normal organs was identified [11]. After L-BPA was injected and well distributed, the patients were irradiated with thermal neutrons and epithermal neutrons produced by a nuclear reactor. ^10^B in the boron-containing drug collided with the high-energy neutron, then underwent a neutron fission reaction, producing a recoiling Li nucleus and an alpha particle and depositing 2.33 MeV of energy within a distance of 5–9 µm. Using this method, normal brain tissue can be spared from radiation damage, giving the patient a second chance at receiving radiotherapy with minimal side effects.

In this study, we enrolled four patients with recurrent meningiomas who had undergone previous craniotomy and radiotherapy, for whom no other effective treatment modalities could be performed. This study aimed to provide an effective salvage treatment option for patients with recurrent meningioma with a low probability of adverse events [12,13].

## 2. Results

The responses after BNCT were collected and are listed in Table 1. The adverse effects of BNCT are tolerable.

### 2.1. Case 115

A 74-year-old man with a skull base meningioma developed neurological symptoms as the disease progressed, including dizziness and bilateral lower leg weakness, and was wheelchair-bound for daily activity (Figure 1). For deep-seated lesions, bilateral irradiation was performed to reduce the maximum dose to the brain while maintaining a sufficient radiation dose to the primary tumor. The total irradiation time increased as the neutron flux decreased with depth. BNCT was completed on 21 August 2020, with a total irradiation time of 1893 s. The tumor average and minimum dose were 14.91 gray equivalent (GyE) and 11.53 GyE, respectively, and the brain maximum and mean dose were 8.34 and 2.96 GyE, respectively. The patient was able to walk independently after undergoing BNCT in August 2020 and reported an improvement in hearing function and tinnitus after BNCT. Magnetic Resonance Imaging (MRI) after BNCT showed stable disease with no obvious interval volume change in the gross tumor (Figure 2). However, a sudden episode of mental deterioration occurred three months after BNCT, with lethargy and poor response to stimulation. The symptoms improved after prednisolone prescription, which was followed up until 28 June 2022. Unfortunately, the patient died of hepatocellular carcinoma with decompensated acute deterioration of liver function.

### 2.2. Case 191

A 45-year-old male with atypical meningioma presented with initial symptoms of left-sided weakness for seven years. He had received four craniotomies and one course of radiotherapy before he came to our hospital. However, he started to develop blurred vision, and rapid tumor growth was also observed in the imaging study. At our hospital, total blindness in both eyes was observed without other cranial nerve deficits. The BNCT was performed on 12 January 2022, with an irradiation time of approximately 910 s. The tumor average dose was 39.15 GyE and the tumor minimum dose was 18.99 GyE, with a brain maximum dose of 11.59 GyE. After irradiation, the patient continued follow-up at our hospital, with the image taken on 15 April 2022 demonstrating significant shrinkage of the tumor (Figure 3). As the tumor showed a promising response to the first BNCT, a second course of BNCT was planned to further control the tumor size. The tumor was treated under the same constraints as the first course and completed on 2022/6/15. No adverse effects or BNCT-related discomforts were reported by the patient.

### 2.3. Case 197

A 58-year-old female was diagnosed with recurrent atypical meningioma, even after craniotomy and gamma-knife radiotherapy. BNCT was completed on 11 February 2022, with an irradiation time of 1428 s, a tumor mean dose of 40.84 GyE, and a minimum dose of 17.62 GyE. The maximum brain dose received was 13.00 GyE. Post-BNCT imaging on 17 March 2022 showed a complete response on the contrast image and a significant reduction in the perifocal edema was observed in the T2 weighted image, which indicates a perfect response for the treatment effect caused by the initial tumor (Figure 4).

### 2.4. Case 284

A 69-year-old male was diagnosed with recurrent spheno-cavernous atypical meningioma after craniotomy and cyber-knife radiotherapy. BNCT was completed on 2022/12/28, with a total irradiation time of 2369 s, a tumor mean dose of 16.83 GyE, and a minimum dose of 9.61 GyE. The maximum brain dose received was 11.29 GyE. Post-BNCT imaging on 30 January 2023 showed stable tumor size on the contrast image, and partial contrast enhancement regression was observed (Figure 5).

In our cohort, the mean tumor dose was 29.414 GyE and the average minimum tumor dose was 15.342 GyE. For normal organs, the average maximum dose received by a normal brain was 11.11 GyE, and the ipsilateral carotid artery mean dose in the first patient was 3.40 GyE. The mean progression-free survival was 14.7 months, with two of the three patients in this cohort showing a response to BNCT. The response rate of BNCT for recurrent atypical meningiomas was 67%.

## 3. Discussion

Meningiomas are the most frequently diagnosed primary central nervous system tumors. Whereas most meningiomas are classified as grade 1 or benign tumors, 15% of these patients developed grade 2 atypical meningiomas and 2% were further identified as having the more aggressive grade 3 malignant meningiomas. The most common symptoms of progressive meningiomas include mass effects, with local invasion causing neurological symptoms and even increasing intracranial pressure, which might be life-threatening if not properly managed [14]. Distant metastasis with the origin of intracranial meningioma is very rare and has only been reported in some case reports of metastasis to the lung or bones [15]. In the whole meningioma population, females have a 2–3 times higher incidence rate than males. However, meningiomas of grade 2 or higher are more common in male patients. Previous studies have also shown that meningiomas of larger size in males and tumors located in the convexity/falx/parasagittal regions have a higher incidence rate of WHO grade 2 or higher [16,17]. In our cohort, we also found that meningiomas located in the high frontal falx area showed more aggressive behavior. Currently, the major treatment modalities for meningiomas are observation and symptomatic control, surgery, hydroxyurea administration, and radiotherapy. Surgery might be the preferred treatment option when symptoms progress, but there are several conditions in which surgery might not be possible, including deep-seated tumors, which might result in severe complications or difficulty achieving complete resection. Simpson grading is a classification system based on surgical resection that also predicts the 10-year recurrence rate [18]. For patients for whom surgery may not be the primary treatment option, radiotherapy may play an important role in tumor control. Radiotherapy techniques include stereotactic radiosurgery for small lesions and conventional linear accelerated radiotherapy, both of which have shown promising control rates in the treatment of meningiomas [19]. Unfortunately, radiotherapy is usually difficult to perform repeatedly, as the accumulated dose limitation of each critical normal organ is limited. For instance, high doses to the brain parenchyma might result in not only cognitive or neurological deficits but also irreversible brain necrosis, which might continue to progress and possibly make surgical debridement necessary. Hence, for patients with meningioma recurrence after a history of receiving radiotherapy, BNCT plays a critical role in the salvage treatment decision. This is because of the biological and physical advantages of BNCT, in which the dose distribution focuses on tumors that uptake boron-10-containing drugs. With this approach, many patients’ meningioma was successfully controlled, leading to an improvement in their quality of life after treatment. Currently, there is an increasing number of neurosurgeons involved in BNCT treatment and an increasing number of patients are referred to the radiation oncology department for the evaluation of the BNCT option.

Fluorodeoxyglucose-positron emission tomography (FDG-PET) retains increased tracer uptake in normal brain tissue, limiting the value of FDG-PET in the interpretation of brain tumors. In contrast, F-fluoro-phenylalanine-positron emission tomography (FBPA-PET) can specifically determine the tumor position [20]. This is due to the difference in the concentration of boron-containing drugs between the tumor and normal brain tissue [21]. The most accepted explanation for this concentration difference is the variety of transporters known as LAT-1, which is an amino acid transporter located on the cell membrane of malignant tumors. As boron-containing drugs have a chemical structure similar to that of tyrosine, they can be transported through LAT-1 transporters, especially in tumor cells that are in a hypermetabolic state [22].

Based on our previous experience, the uptake of boron-containing drugs—confirmed by FBPA-PET—is usually related to the histological aggressiveness of the tumor. For brain tumors, our data showed an average tumor-to-normal tissue (TN) ratio of approximately 2.37, which tends to have a positive correlation with the ki-67 labeling index. Surprisingly, although meningiomas are regarded as slow-growing brain tumors, the TN ratio seems to be satisfactory for BNCT treatment [23]. One reason might be the behavior of recurrent tumors, as residual cancer cells tend to be more resistant to previous treatments and have an accelerated growth pattern [24]. Another reason might be the malignant transformation of the irradiated tumor, with some high-grade components in the recurrent tumor site.

Another PET tracer that is viable for evaluating the distribution of boron-containing drugs is ^18^F-1-amino-3-fluorocyclobutane-1-carboxylic acid (^18^F-fluciclovine) [25]. This drug was initially used for the detection of prostate cancer and its potential spread. In some circumstances, the yield of ^18^F-BPA is limited, with only three patients per week being able to undergo FBPA-PET evaluation. In previous studies, ^18^F-fluciclovine and ^18^F-BPA were shown to share the same amino acid transporter, LAT-1, which theoretically represents a similar distribution of the two drugs. However, ^18^F-fluciclovine is also transported by another system, named sodium-dependent alanine-serine-cysteine transporter 2 (ASCT2). In our single-institute observation, the average TN ratio produced by ^18^F-fluciclovine PET is around two times that of FBPA-PET. This ratio is only an estimation based on experience, as no single patient received both tracers for evaluation and tumor heterogeneity between patients limits the precision of this value. For patient 4, the tumor status and boron uptake were evaluated by ^18^F-fluciclovine PET.

This study had some limitations. First, as malignant meningiomas are relatively rare and BNCT is a developing treatment option, not many cases were included in our study cohort. This results in a lack of significance in our study, as tumor heterogeneity among patients is high in malignancies. Currently, the largest study is a Japanese study that recruited 33 patients with recurrent high-grade meningiomas treated with BNCT and showed a promising treatment effect, with a median survival time of 24.6 months after BNCT [26]. Another limitation of this study is the lack of evidence regarding the precise treatment dose in BNCT to be administered for meningiomas to achieve a satisfactory control rate. In this study, we prescribed the same dose limitation as malignant gliomas, in which the tumor optimally receives 20–40 GyE, which varies based on the radiation dose limitation of adjacent normal tissue tolerance. More experience and data on BNCT for treating meningiomas should be accumulated to determine the tumor dose prescription. The third limitation of this study is the short follow-up time; since meningiomas have relatively slow growing behavior compared to malignant gliomas, a longer follow-up interval may be required to recognize the actual effect of BNCT.

Based on the successful treatment of meningiomas with BNCT, other refractory low-grade malignancies or benign tumors may also be considered potential candidates for BNCT [27]. In the Taipei Veterans General Hospital and Tsing Hua Open-pool Reactor (THOR) collaboration, we have treated patients with malignant peripheral nerve sheath tumor (MPNST) or papillary thyroid carcinoma and have an ongoing trial of ameloblastoma [28]. Some slow-growing tumors share similar characteristics, including causing severe symptoms or neurologic deficits related to tumor compression, a high recurrence rate due to difficulty in gross tumor total resection, and a lack of effective treatment modalities other than surgery for local tumor control. Further studies and trials will be initiated, and we expect more evidence to support the treatment of these tumors using BNCT.

In summary, in this study, we demonstrated the efficacy of salvage BNCT treatment for recurrent meningiomas. By utilizing BNCT, patients with no effective conventional treatment modalities will have the opportunity to experience progression-free survival of malignant meningioma while conserving a good quality of life.

## 4. Materials and Methods

### 4.1. Patients

In this article, we introduce four BNCT cases treated for five courses in Taiwan. All patients were diagnosed with recurrent meningioma. The overall data of the treatment participants are summarized in Table 2, and individual patient characteristics are noted in Table 3. One patient was diagnosed with meningioma (WHO grade 1) and the other three were diagnosed with atypical meningioma (WHO grade 2). All patients received previous primary treatment, whether craniotomy or radiotherapy, and experienced recurrence after the primary treatments. All patients were referred to the Radiation Oncology Department of Taipei Veterans General Hospital (TPEVGH) for evaluation of accessibility and received BNCT treatment at the Tsing Hua Open-pool Reactor (THOR). All treatment progress and ethical standards were approved by the Institutional Review Board (IRB) of TPEVGH and Taiwan’s Food and Drug Administration (TFDA). The WHO grading was described by David N. Louis in “The 2021 classification of Tumors of the Central Nervous System”. The case numbers in the left column of Table 3 represent the serial number of patients receiving compassionate BNCT treatment in Taiwan. Here we introduce the detailed history of the three representative cases.

Case 115, a 74-year-old gentleman with a diagnosis of skull base meningioma, initially presented symptoms of dizziness and intermittent headache in 2018. MRI showed a skull base tumor. The symptoms progressed because the patient refused to undergo craniotomy due to bilateral lower limb weakness and unsteady gait. Repeated MRI during 2020 showed the tumor to be 59.6 × 48.1 × 54.1 mm in size, involving the right part of the tentorium, Meckel’s cave, the cavernous sinus, and the right suprasellar region. Encasement of the major arteries around the circle of Willis and the basilar artery were also noted, with severe compression of the ventral aspect of the brainstem causing obstructive hydrocephalus. Hydroxycarbamide was prescribed and radiotherapy was performed at a dose of 2800 cGy in 14 fractions. Follow-up imaging after the primary treatment showed stable disease and further treatment with BNCT was performed. BNCT was completed on 2020/8/21.

Case 197, a 58-year-old patient, was diagnosed with an atypical meningioma. She was diagnosed in December 2011 with an initial presentation of intermittent headache. A craniotomy for tumor removal was performed in 2011 and it proved to be an atypical meningioma. Gamma knife stereotactic radiosurgery for recurrent atypical meningioma was performed between July 2013 and April 2019. During follow-up, a brain MRI in January 2021 showed recurrent meningioma, and the patient underwent a second craniotomy in March 2021. The pathology was similar to that of a previous craniotomy, which proved to be an atypical meningioma without obvious malignant transformation or further upgrading of tumor aggressiveness. Unfortunately, recurrence over the vertex was discovered in September 2021, with a size of 27 × 38 mm; thus, radiotherapy with 4000 cGy in 20 fractions was performed for the recurrent lesion. As the imaging study after radiotherapy showed residual tumor progression, BNCT was suggested for further local tumor control. BNCT was completed on 11 February 2022.

Case 284, a 69-year-old patient, was diagnosed with an atypical meningioma of the right spheno-cavernous sinus. The tumor was discovered during a regular health examination in America in January 2018, and the patient returned to Taiwan for a meningioma craniotomy. Recurrence was noticed in February 2019 and radiotherapy with CyberKnife was performed for local tumor control. However, further imaging follow-ups in 2020 showed tumor progression, and the neurological symptoms deteriorated to ptosis and oculomotor nerve (CN3) compression. A repeat craniotomy was performed in August 2020. Unfortunately, the status of the patient continued to worsen, with impaired memory, unsteady gait, and intermittent urine incontinence. In the imaging study in August 2022, tumor progression over the right orbital apex and right intraconal space was observed, as well as extension to the bilateral cavernous sinus. A repeat craniotomy for decompression was then performed. However, a persistent residual tumor was still observed on the postoperative MRI. We were consulted for salvage BNCT treatment to control the aggressive meningioma. The BNCT was completed on 28 December 2022.

### 4.2. Treatment Protocol

All patients in this study underwent 4-borono-2-^18^F-fluoro-phenylalanine (FBPA) or ^18^F-Fluciclovine positron emission tomography as a tool for evaluating the efficacy of BNCT treatment. The advantages of FBPA and ^18^F-Fluciclovine PET over conventional fluorodeoxyglucose (FDG) PET are demonstrated in the discussion section. After calculating the standardized uptake value (SUV) of the tumor and the normal brain (usually the contralateral cerebellum), the tumor-to-normal tissue ratio was determined. Based on previous studies, a TN ratio of FBPA-PET greater than 2.5 indicates adequate tumor uptake, where the dose gradient is good enough to transfer into a treatment benefit over the damage to the normal brain [29]. For patients undergoing FBPA-PET, the TN ratio was integrated into the treatment planning system to evaluate the radiation dose to the tumor and normal tissue. As for patients who arranged ^18^F-Fluciclovine PET to evaluate BNCT indication, the same threshold of 2.5 was used to determine the treatment probability; however, a constant value of 2.5 was integrated into the treatment planning system, as ^18^F-Fluciclovine PET might overestimate the exact concentration of boron-containing drug uptake by the tumor.

The neutron beam produced by the reactor includes several types of radiation beams, including gamma rays and X-rays [30]. The total radiation dose was summed by the treatment planning system with the unit named gray equivalent (GyE), defined as the sum of all types of radiation using the Monte Carlo method [31,32]. The dose limitation for 1% of the normal brain volume and the mean brain dose were 12 GyE and 3 GyE, respectively. The tumor mean dose is optimal in the range of 20–40 GyE but is constrained by the dosage to the normal tissue dose limitation [33].

On the day of BNCT, all patients received a continuous infusion of L-BPA 2 h before irradiation, followed by half the infusion rate during the whole treatment duration. The infusion protocol for total L-BPA was 500 mg/kg of body weight. In other words, the patient received a continuous infusion of 200 mg/kg of body weight per hour for 2 h, followed by 100 mg/kg of body weight per hour during the entire neutron irradiation period. For the four patients, the blood boron concentrations were 34.11 ppm, 35.35 ppm, 27.23 ppm, and 32.22 ppm. Before the BNCT treatment was conducted, the treatment planning system showed the dose of each structure under blood boron concentrations of 25 ppm and 35 ppm. It was surveyed in a previous study that most patients’ blood boron concentrations were in the range of 25 to 35 ppm with the previously mentioned infusion protocol.

After BNCT was completed, the patient received monthly bevacizumab to reduce brain edema caused by irradiation for at least two months. Follow-up MRI was scheduled at least one month after BNCT was performed [34].

### 4.3. Response Evaluation

The response after BNCT was evaluated using Response Evaluation Criteria in Solid Tumors (RECIST), categorizing treatment outcomes into complete response (CR), partial response (PR), stable disease (SD), or progressive disease (PD) [35]. The patients underwent an MRI study approximately four weeks after BNCT treatment to minimize misleading tumor response caused by radiation-related brain edema. Progression-free survival was defined as the date from the end of BNCT to the date of the last follow-up and was only evaluated for patients who completed BNCT for more than three months with a post-BNCT imaging follow-up. The final survival analysis was completed on 31 December 2022.

## Figures and Tables

**Figure 1 ijms-24-04978-f001:**
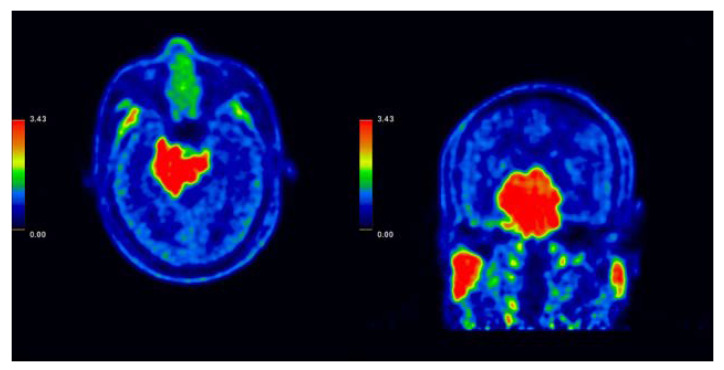
(**Left**) F-fluoro-phenylalanine-positron emission tomography (FBPA-PET) axial image of case 115, with the tumor-to-normal tissue standardized uptake value (SUV) average calculated after radioactive boron-containing drugs were infused into the patient. Other than the tumor, physiological uptake was also noticed in the parotid glands. The ratio of this patient is 2.89, matching the requirement of BNCT treatment (T/N ratio > 2.5) (**Right**) FBPA-PET coronal image.

**Figure 2 ijms-24-04978-f002:**
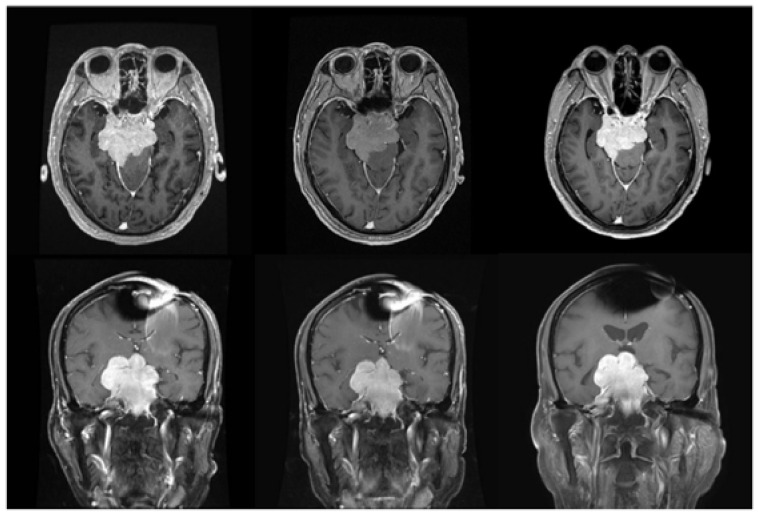
Magnetic Resonance Imaging (MRI) image of case 115 with skull base meningioma. (**Left upper**) Pre-BNCT, axial MRI with contrast. (**Left lower**) Pre-BNCT, coronal MRI with contrast. (**Middle upper**) Three months after BNCT, axial MRI with contrast. (**Middle lower**) Three months after BNCT, coronal MRI with contrast. (**Right upper**) 17 months after BNCT, axial MRI with contrast (**Right lower**) 17 months after BNCT, coronal MRI with contrast.

**Figure 3 ijms-24-04978-f003:**
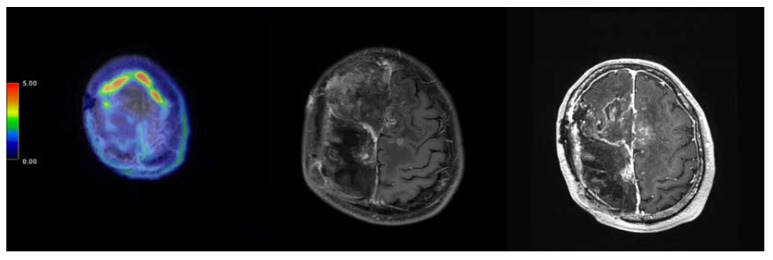
MRI image of case 191 with skull base meningioma. (**Left**) Pre-BNCT, FBPA-PET image. (**Middle**) Pre-BNCT, axial MRI with contrast. (**Right**) Three months after BNCT, axial MRI with contrast.

**Figure 4 ijms-24-04978-f004:**
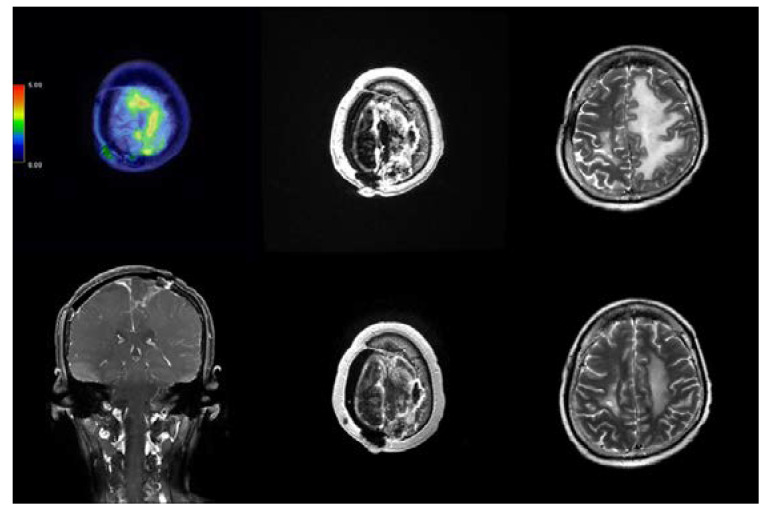
Images of case 197 with left high frontal meningioma. (**Upper left**) Pre-BNCT, FBPA-PET image, with tumor-to-normal tissue (TN) ratio around 2.64. (**Upper middle**) Pre-BNCT, axial MRI with contrast. (**Upper right**) Pre-BNCT, axial MRI T2 weighted image. (**Lower left**) Two months after BNCT, coronal MRI with contrast. (**Lower middle**) Two months after BNCT, axial MRI with contrast. (**Lower right**) Two months after BNCT, axial MRI T2 weighted image.

**Figure 5 ijms-24-04978-f005:**
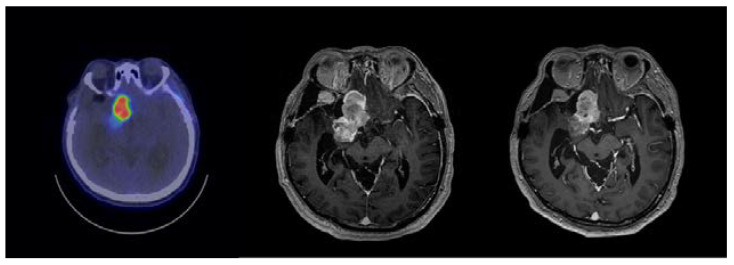
Images of case 284 with recurrent spheno-cavernous atypical meningioma. (**Left**) Pre-BNCT, Fluciclovine-PET image, with tumor-to-normal tissue (TN) ratio around 7.07. (**Middle**) Pre-BNCT, axial MRI with contrast. (**Right**) Post-BNCT, axial MRI with contrast two months after BNCT.

**Table 1 ijms-24-04978-t001:** Final dose report of Boron Neutron Capture Therapy (BNCT) treatment and the treatment response categorized by the Response Evaluation Criteria in Solid Tumors (RECIST) criteria.

Case	BNCT Date	Irradiation Time (Sec)	Tumor Average Dose (GyE)	Tumor Minimum Dose (GyE)	Brain Maximum Dose (GyE)	Tumor Response	Last Follow-Up	PFS (Months)
115	21 August 2020	1893	14.91	11.53	8.34	SD	28 June 2022, SD	23
191	12 January 2022 15 June 2022	910 1055	39.15 35.34	18.99 18.96	11.59 12.48	PR SD	13 December 2022, PR	11
197	11 February 2022	1428	40.84	17.62	13.00	CR	20 December 2022, CR	10
284	28 December 2022	2369	16.83	9.61	11.29	SD	10 January 2023	2

SD: stable disease; PR: partial response; CR: complete response; PFS: progression-free survival.

**Table 2 ijms-24-04978-t002:** General patient characteristics. M: male; F: female; Conc.: concentration; GyE: Gray equivalent; BED_6_: biological effective dose with α/β = 6.

	All Patients (*n* = 4)
Age (years)	61.75
Gender (M, F)	3, 1
Mean T/N ratio	4.125
Mean irradiation time (sec)	1531
Mean blood boron conc. (ppm)	32.23
Mean tumor average dose (GyE)	29.414
Mean tumor BED_6_	194.58
Mean tumor minimum dose (GyE)	15.342
Mean brain max dose (GyE)	11.11

**Table 3 ijms-24-04978-t003:** Patient characteristics, including the pathology and the tumor site. TN ratio: tumor-to-normal tissue ratio; RT: radiotherapy; ChemoT: chemotherapy; OP: operation; SC sinus: spheno-cavernous sinus.

Case	Age/Gender	Initial Pathology	Site	TN Ratio	Previous Treatment
115	74/Male	Meningioma	Skull base	2.89	RT + ChemoT
191	45/Male	Atypical meningioma	Right high frontal	3.9	RT + OP + ChemoT
197	59/Female	Atypical meningioma	Right high frontal	2.64	RT + OP + ChemoT
284	69/Male	Atypical meningioma	Right SC sinus	7.07	RT + OP + ChemoT

## Data Availability

The data are not publicly available due to patient privacy.

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
