# Peer review of "Advances in Boron Neutron Capture Therapy (BNCT) for Recurrent Intracranial Meningioma"

_ijms, 2023, doi:10.3390/ijms24054978_

Round 1

Reviewer 1 Report

Tian-Li Lan et al provide a case series of 4 patients with meningioma, all with previous surgery and radiation therapy, who were provided with Boron neutron capture therapy.  The authors describe responses in 3 of the 4 cases; one (case 284) did not have any information regarding demographics or response.

Abstract: Accurately summarizes the report findings.

Introduction: Adequately describes the hypothesis and background.

Results: Describes responses in 3 of the 4 patients.   The response in case 284 needs to be added.

Materials and Methods: Adequately describes patient tumor grade, demographics, previous treatment, BNCT treatment details and assessment of response.  Case descriptions are described in the results section (115, 191 and 197) and in the materials and methods section (115, 197 and 284).  These should al be consolidated into the results section.

Discussion: Adequately discusses the findings, the rationale for the responses, study limitations and translation to future studies and trials.

Author Response

Dear reviewer,

Thank you for your precious suggestions. We added the radiographic response of case 284, which was followed recently. In the “Materials and Methods”, we aim to show the previous medical managements the patient received, which usually include surgery and irradiation. These procedures make the patient extremely difficult to receive other treatments. In “Results”, we will like to show more statistic data about the current BNCT treatment, covering the irradiated dose and treatment responses. Thank you so much for the advice to make this article more comprehensive.

Reviewer 2 Report

I have several minor comments. Abstract and the beginning of the introduction part are completely the same. I suppose, author should rewrite the abstract part and give more details of present research. In addition, authors should make full-fledged conclusions with the indication of main results of present work.

Author Response

Dear reviewer,

Thank you for your precious comment. We have re-wrote the Abstract, and also included the conclusions of this article. Thank you for the advice to make our abstract more comprehensive.

Reviewer 3 Report

Authors present a case series on 4 patients with malignant cranial meningiomas who were, following craniotomy and radiotherapy, treated with Boron Neutron Capture Therapy (BNCT). Although accentuation of this manuscript is on the radiation therapy, I suggest to include for all 4 cases initial MRIs and describe the surgical therapy as well as all therapy modalities prior to BNCT - also clinical aspect of the patients is missing (deficits, complications etc). We also need to know for all these patients - when was the initial diagnosis, when were surgeries/radiotherapy and in which technique, what was the time span, how long is the follow up. This is all included in Materials and MEthods, but I suggest to include this - with initial scans - in the Results section. Did you have any cases of radiation necrosis? 

Most important - what was the histology of these tumors? Overall survival? Follow up?

I suggest to re-write the manuscript as tehnical note, since there are larger series on this subject which authors do not cite, so I suggest to cite and include:

Takeuchi K, Kawabata S, Hiramatsu R, Matsushita Y, Tanaka H, Sakurai Y, Suzuki M, Ono K, Miyatake SI, Kuroiwa T. Boron Neutron Capture Therapy for High-Grade Skull-Base Meningioma. J Neurol Surg B Skull Base. 2018 Oct;79(Suppl 4):S322-S327. doi: 10.1055/s-0038-1666837. Epub 2018 Jul 3. PMID: 30210985; PMCID: PMC6133692.

Miyatake SI, Wanibuchi M, Hu N, Ono K. Boron neutron capture therapy for malignant brain tumors. J Neurooncol. 2020 Aug;149(1):1-11. doi: 10.1007/s11060-020-03586-6. Epub 2020 Jul 16. PMID: 32676954.

Author Response

Dear reviewer,

Thank you for your precious comments. As BNCT was performed for recurrent meningiomas for salvage intention, with patients in this study received their initial treatment in other hospitals or even other countries, we tried to get as much information as possible. We looked for medical records that are uploaded to the National Health Insurance system in Taiwan, and also asked the patient to provide previous treatment results, that might influence the treatment.

Regarding radiation necrosis, as previous studies showed the advantage of adding bevacizumab after BNCT, we infuse bevacizumab on the day after BNCT was arranged, and monthly bevacizumab after BNCT. No radiation necrosis was mentioned in the image study of this four patients. We also provided the histology in Table 3, and the progression free survival in Table 1. Regarding the overall survival, as 3 of the patients are still alive, we will need a longer follow up time to provide the survival analysis. We have also reviewed the two articles you mentioned, that have very encouraging and mesmerizing results. We added this two articles as our reference 13 and reference 26. Thank you so much for your advice to make this article more comprehensive.

Round 2

Reviewer 1 Report

All prior comments were satisfactorily addressed

Reviewer 3 Report

Authors have sufficiently responded to reviewer remarks.